# Bayesian Treatment of the Spectrum of the Empirical Kernel in (Sub)Linear-Width Neural Networks

**Ouns El Harzli**
Department of Computer Science
University of Oxford
Oxford, UK
ouns.elharzli@new.ox.ac.uk

**Bernardo Cuenca Grau**
Department of Computer Science
University of Oxford
Oxford, UK
bernardo.cuenca.grau@cs.ox.ac.uk

## ABSTRACT

We study Bayesian neural networks (BNNs) in the theoretical limits of infinitely increasing number of training examples, network width and input space dimension. Our findings establish new bridges between kernel-theoretic approaches and techniques derived from statistical mechanics through the correspondence between Mercer's eigenvalues and limiting spectral distributions of covariance matrices studied in random matrix theory. Our theoretical contributions first consist in novel integral formulas that accurately describe the predictors of BNNs in the asymptotic linear-width and sublinear-width regimes. Moreover, we extend the recently developed renormalisation theory of deep linear neural networks, enabling a rigorous explanation of the mounting empirical evidence that hints at the theory's applicability to nonlinear BNNs with ReLU activations in the linear-width regime. From a practical standpoint, our results introduce a novel technique for estimating the predictor statistics of a trained BNN that is applicable to the sublinear-width regime where the predictions of the renormalisation theory are inaccurate.

## 1 INTRODUCTION

Bayesian Neural Networks (BNNs) are a variant of neural networks that incorporate Bayesian inference techniques to mitigate overfitting, enable learning from small datasets, and capture uncertainty in predictions (Neal, 2012; Gal, 2016). In a BNN, prior probability distributions are specified for weights and biases. During training, the posterior distribution, which represents the updated knowledge about the parameters after observing the data, is updated using Bayes' rule. A trained BNN can be interpreted as an infinite ensemble of neural networks where each individual contribution in the ensemble is weighted by the posterior probability of its parameters given the training data. Although computing the posterior distribution is intractable and difficult to approximate, BNNs have gained significant traction with the development of effective estimation techniques (Gal, 2016; Blei et al., 2017).BNNs demonstrate generalisation performance on par with deep neural networks trained using gradient descent (Lee et al., 2020; Magris & Iosifidis, 2023). BNNs also showcase improved sensitivity to out-of-distribution examples (Gal, 2016) and the ability to estimate uncertainty.

In an effort to analyse the generalisation properties of BNNs, researchers study idealised views of fully-connected neural architectures defined by the input dimension, the layer widths, and the activation function. As the width approaches infinity in each layer (the *NNGP limit*), the functions generated by random weight selection converge in distribution to a Gaussian process (GP) (Rasmussen & Williams, 2006). The covariance function of such GP, called the *NNGP kernel*, can be recursively defined by proceeding on a layer by layer basis (Lee et al., 2018). This perspective based on kernel and GP theory has inspired formalisms that mimic different aspects of the behavior of BNNs in the infinite-width limit Aitchison et al. (2021), including representation learning Yang et al. (2023). Simultaneously, it has led to the development of analytical formulas to estimate the generalisation error of related kernel and random features models (Canatar et al., 2021; Simon et al., 2023). These formulas often rely on the *spectral universality assumption* (SUA), which simplifies the derivations by approximating the eigenfunctions of the kernel with independent Gaussian entries

(Karoui, 2010; Cheng & Singer, 2013; Fan & Montanari, 2015). Extensive research is being devoted to study the accuracy of the SUA (Liu et al., 2021; Lu & Yau, 2023; Bosch et al., 2023).

In addition to the NNGP limit, BNNs have also been studied under the *linear-width limit* (also referred to as *thermodynamic limit* or *proportional limit*) where the network's width, the number of training examples and the dimension of the input space are taken simulataneously to infinity while keeping constant and bounded ratios between them (Engel et al., 2012). By employing techniques from statistical mechanics, such as saddle point approximations (Seung & Sompolinsky, 1992; Li & Sompolinsky, 2021), the replica method (Barbier et al., 2018; Canatar et al., 2021), and random matrix theory (Wigner, 1955; Livan et al., 2018; Fan & Wang, 2020), researchers have studied the mean and variance of the output generated by trained BNNs in this setting. A recent theoretical work (Cui et al., 2023) has derived the predictor learned by non-linear BNNs in the case of Gaussian data. More recently, *sublinear-width regimes*, where the width (or the input dimension) is small compared to the number of data points (Maillard et al., 2024), and related scalings (van Meegen & Sompolinsky, 2024) have been studied, and the emergence of strong feature learning has been demonstrated in these scenarios.

One of the most prominent results in this literature is the *renormalisation theory* (Li & Sompolinsky, 2021) of linear BNNs (i.e., those without non-linear activations) in the linear-width regime, which establishes that the mean predictor and the predictor variance of the BNN coincide with that of Bayesian linear regression, but surprisingly the variance must be renormalised by a factor dependent on the training data and problem dimensions. Subsequent developments have provided more detailed analysis on the linear setting including non-asymptotic results (Hanin & Zlokapa, 2023), and comparison with deep random feature models (Zavatone-Veth et al., 2022). It remains an open question, however, whether the insights from the renormalisation theory for linear BNNs can be extended to non-linear networks, as suggested by empirical evidence (Li & Sompolinsky, 2021; Ariosto et al., 2023), and how the theory should be adapted to sublinear-width regimes, where discrepancies with empirical results have been observed (Li & Sompolinsky, 2021).

**Our Contributions** In this paper, we establish new connections between the kernel-theoretic perspective associated with the NNGP limit and the statistical mechanics viewpoint associated with the linear-width and sublinear-width limits, and contribute new insights to the generalisation properties of BNNs. First, we demonstrate that training a (non-linear) BNN in the linear-width and sublinear-width limits result in a predictor with identical mean and variance to that of GP regression with a modified NNGP kernel, and we observe that the Mercer spectrum (Mercer, 1909; Minh et al., 2006) of this kernel is known in the linear-width regime. Second, using this observation, we prove necessary and sufficient conditions (on the data and the architecture) for the application of renormalisation theory to non-linear BNNs in the linear-width limit. These conditions also provide a criterion for determining the applicability of the spectral universality assumption (SUA) from kernel theory in the context of BNNs. Third, we present initial findings on a sublinear-width regime where the relevant quantities are simultaneously taken to infinity while the number of training examples remains proportional to the product of the network width and the dimension of the input space. In particular, we provide a novel mechanism for estimating the mean and variance of the predictions of non-linear BNNs in this setting, for which renormalisation theory is not applicable.

## 2 PRELIMINARIES

We use standard notation for real-valued vectors $\mathbf{v} \in \mathbb{R}^n$, matrices $\mathbf{A} \in \mathbb{R}^{m \times n}$, and their transposes $\mathbf{v}^T$ and $\mathbf{A}^T$. We use $\mathbf{a}_i$ to denote the vector in the $i$-th row of $\mathbf{A}$. The Moore-Penrose pseudo-inverse of a matrix $\mathbf{A}$ is denoted as $\mathbf{A}^\dagger$ (Moore, 1920).

**Neural networks.** A fully-connected neural (FCN) architecture with $L$ layers is a tuple $f = \langle \{\mathbf{W}^\ell\}_{1 \le \ell \le L}, \{\mathbf{b}^\ell\}_{1 \le \ell \le L}, \{\sigma^\ell\}_{1 \le \ell \le L} \rangle$. Each layer $\ell \in \{1, ..., L\}$ of width $N_\ell$ is given by a weight matrix $\mathbf{W}^\ell \in \mathbb{R}^{N_\ell \times N_{\ell-1}}$, a bias $\mathbf{b}^\ell \in \mathbb{R}^{N_\ell}$ and an activation function $\sigma^\ell : \mathbb{R} \mapsto \mathbb{R}$. On input $\mathbf{x} \in \mathbb{R}^{N_0}$, network $f$ sets $\mathbf{x}^0 = \mathbf{x}$ and then computes recursively on the depth the sequence of pre-activations $\mathbf{h}^\ell$ and activations $\mathbf{x}^\ell$ as follows, where the network's output $f(\mathbf{x})$ is given by $\mathbf{x}^L$:

$$\mathbf{h}^\ell = \mathbf{W}^\ell \cdot \mathbf{x}^{\ell-1} + \mathbf{b}^\ell \qquad \mathbf{x}^\ell = \sigma^\ell(\mathbf{h}^\ell) \tag{1}$$

We assume that all but the last layer have the same width $N$. For the last layer, we assume width $N_L = 1$ (ensuring a real-valued output), $b^L = 0$ and $\sigma^L = \mathrm{Id}_{\mathbb{R}}$ (ensuring linearity). In this setting, the weights $\mathbf{W}^L$ are referred to as the readout weights (Li & Sompolinsky, 2021).

**Kernels.** A kernel on $\mathbb{R}^{N_0}$ is a positive semi-definite symmetric function $K : \mathbb{R}^{N_0} \times \mathbb{R}^{N_0} \mapsto \mathbb{R}$. By Mercer's theorem (Minh et al., 2006), given a distribution $\mathbf{x} \sim p(\mathbf{x})$ with compact support on $\mathbb{R}^{N_0}$, there exist unique countable collections of Mercer's eigenvalues $(\lambda_i)_{i \in \mathbb{N}}$ and eigenfunctions $(\varphi_i)_{i \in \mathbb{N}}$ such that $K(\mathbf{x}, \mathbf{x}') = \sum_i^\infty \lambda_i \varphi_i(\mathbf{x}) \varphi_i(\mathbf{x}')$ and $(\varphi_i)$ are orthonormal w.r.t. the data distribution: $\mathbb{E}_{\mathbf{x} \sim p(\mathbf{x})} (\varphi_i(\mathbf{x}) \varphi_j(\mathbf{x})) = \delta_{i,j}$ for all $i, j$. By Riesz's theorem, there exists a Hilbert space $\mathcal{H}$ and a feature map $\phi : \mathbb{R}^{N_0} \mapsto \mathcal{H}$ such that $K(\mathbf{x}, \mathbf{x}') = \langle \phi(\mathbf{x}), \phi(\mathbf{x}') \rangle_{\mathcal{H}}$. Kernel regression amounts to linear regression in the corresponding Hilbert space: when trained on data $\mathbf{X}, \mathbf{y}$, the prediction on a new point $\mathbf{x}^*$ is given by $\mathbf{k}_{\mathbf{x}^*, \mathbf{X}}^T \mathbf{K}_{\mathbf{X}, \mathbf{X}}^{-1} \mathbf{y}$ where the vector $\mathbf{k}_{\mathbf{x}^*, \mathbf{X}}$ is given by $(\mathbf{k}_{\mathbf{x}^*, \mathbf{X}})_i = K(\mathbf{x}^*, \mathbf{x}_i)$ and the kernel matrix $\mathbf{K}_{\mathbf{X}, \mathbf{X}}$ is given by $(\mathbf{K}_{\mathbf{X}, \mathbf{X}})_{i,j} = K(\mathbf{x}_i, \mathbf{x}_j)$. Although the kernel's eigenfunctions exhibit the described structure, the *spectral universality assumption* (SUA) is commonly adopted. The SUA posits that, as $P$ increases, the eigenfunctions can be approximated by independent Gaussian entries: $\varphi_i(\mathbf{x}_j) \sim \mathcal{N}(\mu_K, \sigma_K^2)$, where $\mu_K$ and $\sigma_K^2$ depend on the kernel $K$ and the data distribution $p(\mathbf{x})$, but not on specific instances $i$ and $j$. The SUA works well in practice (Karoui, 2010; Cheng & Singer, 2013; Fan & Montanari, 2015; Liu et al., 2021; Simon et al., 2023; Lu & Yau, 2023; Schröder et al., 2023), and research focuses on identifying conditions under which it holds.

**Random feature maps.** Let $\Theta$ represent all parameters of $f$ up to layer $L - 1$. The *random feature map* $\phi(\Theta, \cdot) : \mathbb{R}^{N_0} \mapsto \mathbb{R}^N$ is a nonlinear transformation (random in $\Theta$) mapping the input and the activation $\mathbf{x}^{L-1}$. By definition, $f(\mathbf{x}) = (\mathbf{W}^L)^T \phi(\Theta, \mathbf{x})$, and to highlight the parameter dependency we denote it as $f_{\Theta, \mathbf{W}^L}$. The random feature map is associated to a *random kernel* $K_\Theta^{N, N_0} : (\mathbf{x}, \mathbf{x}') \mapsto \frac{1}{N} \langle \phi(\Theta, \mathbf{x}), \phi(\Theta, \mathbf{x}') \rangle$ expressed as the inner product between the corresponding random feature map evaluations. For this kernel, the Hilbert space $\mathcal{H} = \mathbb{R}^N$ is thus known.

**Training set.** The training set $(\mathbf{X}, \mathbf{y})$ consists of $P$ examples sampled i.i.d. from an unknown distribution $\mathbb{P}_{N_0}$ with compact support on $\mathbb{R}^{N_0} \times \mathbb{R}$. We assume that in the limit $N_0 \to \infty$, $\mathbb{P}_{N_0}$ converges (in distribution) to a well-defined distribution with compact support over $\mathbb{R}^{\mathbb{N}} \times \mathbb{R}$ noted $\lim_{N_0 \to \infty} \mathbb{P}_{N_0}$. We denote each example by $(\mathbf{x}_i, y_i)$, so that $\mathbf{X} = (\mathbf{x}_1, ..., \mathbf{x}_P)^T \in \mathbb{R}^{P \times N_0}$ and $\mathbf{y} = (y_1, ..., y_P)^T \in \mathbb{R}^P$. We denote the evaluation of the random feature map on the training set by $\phi(\Theta, \mathbf{X}) = (\phi(\Theta, \mathbf{x}_1), ..., \phi(\Theta, \mathbf{x}_P))^T \in \mathbb{R}^{N \times P}$; this induces an empirical kernel matrix $\mathbf{K}_\Theta^{P, N, N_0}(\mathbf{X}, \mathbf{X})$ given by $\frac{1}{N} [\phi(\mathbf{X}, \Theta)]^T \phi(\mathbf{X}, \Theta) \in \mathbb{R}^{P \times P}$. The training data $\mathbf{X}$ also induces an empirical distribution $p_{\mathbf{X}}(\mathbf{x}) = \frac{1}{P} \left( \sum_{i=1}^P \delta_{\mathbf{x}_i}(\mathbf{x}) \right)$ with $\delta_{\mathbf{x}_i}$ the Dirac measure.

**BNNs.** We assume a *prior distribution* over parameters $(\Theta, \mathbf{W}^L)$ with weights sampled i.i.d. from $\mathcal{N}(0, \frac{1}{N})$ and biases sampled i.i.d. from $\mathcal{N}(0, 1)$; this yields a density $p(\Theta, \mathbf{W}^L)$ that is a product of Gaussian densities. The *posterior distribution* given the training data is given by Bayes' rule:

$$p(\Theta, \mathbf{W}^L | \mathbf{X}, \mathbf{y}) = p(\Theta, \mathbf{W}^L) \frac{p(\mathbf{y} | \mathbf{X}, \Theta, \mathbf{W}^L)}{p(\mathbf{y} | \mathbf{X})}$$

where $p(\mathbf{y} | \mathbf{X}, \Theta, \mathbf{W}^L)$ is the *likelihood* of the data given a set of parameters, and $p(\mathbf{y} | \mathbf{X}) = \int p(\mathbf{y} | \mathbf{X}, \Theta, \mathbf{W}^L) p(\Theta, \mathbf{W}^L) \mathrm{d}\Theta \mathrm{d}\mathbf{W}^L$. is the *marginal likelihood* (or *evidence*). We assume Gaussian likelihoods, i.e. $p(\mathbf{y} | \mathbf{X}, \Theta, \mathbf{W}^L) \sim \mathcal{N}(\mathbf{y}, \phi(\Theta, \mathbf{X})^T \mathbf{W}^L \mathbf{W}^{L^T} \phi(\Theta, \mathbf{X}))$. Calculating the posterior distribution, which is the essence of BNN training, is analytically intractable and remains a core challenge (Gal, 2016). In practice, the posterior distribution is estimated via variational inference (Blei et al., 2017) or Monte-Carlo simulation methods (Rasmussen, 1995).

Given the posterior distribution, the predictor defines a distribution over functions $f_{\Theta, \mathbf{W}^L}$ with $(\Theta, \mathbf{W}^L) \sim p(\Theta, \mathbf{W}^L | \mathbf{X}, \mathbf{y})$. The mean-squared generalisation error is defined for any new point $(\mathbf{x}^*, y^*)$ as the expectation over the predictor error: $\mathbb{E}_{(\Theta, \mathbf{W}^L) \sim p(\Theta, \mathbf{W}^L | \mathbf{X}, \mathbf{y})} \left( (y^* - f_{\Theta, \mathbf{W}^L}(\mathbf{x}^*))^2 \right)$. Only the mean and variance of the predictor are needed to calculate it.

**Gaussian processes and NNGPs.** A GP $g$ over a space $\mathbb{R}^{N_0}$ is a random scalar field such that its evaluation at any collection of finitely many points $(g(x_1), ..., g(x_P))$ follows a multivariate Gaussian distribution. A GP is determined by a mean function $\mu : \mathbb{R}^{N_0} \mapsto \mathbb{R}$, and a covariance function $K : \mathbb{R}^{N_0} \times \mathbb{R}^{N_0} \mapsto \mathbb{R}$, which describe respectively the mean of the Gaussian distribution at each point and the covariance between the Gaussians at any two points. The covariance function of a GP is a kernel (Rasmussen & Williams, 2006). We note $g \sim \mathcal{GP}(\mu, K)$. *Gaussian process regression* consists in performing Bayesian inference using a Gaussian process as the prior distribution over functions. The prediction distribution of GP regression with prior $\mathcal{GP}(0, K)$ trained on the data $\mathbf{X}, \mathbf{y}$ is given, on a new point $\mathbf{x}^*$, by $\mathcal{N}(\mathbf{k}_{\mathbf{x}^*,\mathbf{X}}^T \mathbf{K}_{\mathbf{X},\mathbf{X}}^{-1} \mathbf{y}, K(\mathbf{x}^*, \mathbf{x}^*) - \mathbf{k}_{\mathbf{x}^*,\mathbf{X}}^T \mathbf{K}_{\mathbf{X},\mathbf{X}}^{-1} \mathbf{k}_{\mathbf{x}^*,\mathbf{X}})$. The mean prediction of GP regression coincides with the prediction of kernel regression with the same kernel.

Applying successively the central limit theorem to each layer, the infinite-width limit of equation 1 yields a GP, called the *Neural Network Gaussian Process (NNGP)*. If we let the width $N \to \infty$, the $\mathbf{h}_i^L \sim \mathcal{GP}(\mu^L, K^L)$ are independent and defined inductively by layers as follows for all $\mathbf{x}, \mathbf{x}' \in \mathbb{R}^{N_0}$ and each $\ell \in 1, ..., L$. First, $\forall \ell$ $\mu^\ell(\mathbf{x}) = 0$ and $K^0(\mathbf{x}, \mathbf{x}') = \mathbf{x}^T \mathbf{x}'$. Then, $\mathbf{h}_i^{\ell-1} \sim \mathcal{GP}(\mu^{\ell-1}, K^{\ell-1})$ and the covariance functions $K^\ell(\mathbf{x}, \mathbf{x}')$ are given by $\mathbb{E}_{\mathbf{h}_i^{\ell-1} \sim \mathcal{GP}(\mu^{\ell-1}, K^{\ell-1})} \left( \sigma^\ell(\mathbf{h}_i^{\ell-1}(\mathbf{x})) \sigma^\ell(\mathbf{h}_i^{\ell-1}(\mathbf{x}')) \right)$. The covariance function $K^L$ is the *NNGP kernel* (Daniely et al., 2016), denoted as $K^L = K_{\mathrm{NNGP}}$. Infinite-width limits involve various subtleties (Matthews et al., 2018), and we follow the approach in Lee et al. (2018) where infinite limits are taken sequentially. In this limit, the number of examples $P$ and the input dimension $N_0$ remain fixed. Furthermore, we will investigate more comprehensive limits where $P$, $N$, and $N_0$ all tend to infinity simultaneously, first while maintaining constant and bounded ratios $\alpha = \frac{P}{N}$ and $\alpha_0 = \frac{P}{N_0}$ (linear-width regime), then while $P \propto N \cdot N_0$, thus $\alpha \to \infty$ and $\alpha_0 \to \infty$ (sublinear-width regime).

**Random matrix theory.** Random matrix theory (Wigner, 1955; Livan et al., 2018) is the study of the spectral distributions of large matrices of random variables. The spectral measure $F_P$ for a given matrix, with eigenvalues $\lambda_i$, is given, for $x \in \mathbb{R}$, by $F_P(x) := \frac{1}{P} \sum_{i=1}^P \delta_{\lambda_i}(x)$, where $\delta_{\lambda_i}(x)$ represents the Dirac measure centered at the eigenvalue $\lambda_i$. When the matrix is random, the spectral measure becomes a random measure, called the empirical spectral distribution. Our focus lies in studying weak convergences (convergences in distribution) of the spectral measures towards nonrandom measures (Geronimo & Hill, 2002). A sufficient condition for weak convergence of measures is to have pointwise convergence in their Stieltjes transforms (Geronimo & Hill, 2002). We rely on a famous result in random matrix theory. Consider $\mathbf{W} \in \mathbb{R}^{N \times P}$, a random matrix with i.i.d. entries drawn from $\mathcal{N}(0, \frac{1}{N})$ and $\mathbf{\Psi}$ a nonrandom positive semi-definite matrix. Suppose that $\mathbf{\Psi}$ has a limiting spectral measure $\rho$, and let $P, N \to \infty$ with fixed ratio $\alpha := \frac{P}{N}$, then the random matrix $\mathbf{\Psi}^{1/2} \mathbf{W}^T \mathbf{W} \mathbf{\Psi}^{1/2}$ has a limiting nonrandom spectral measure $\rho_{MP}^\alpha \boxtimes \rho$. The Marchenko-Pastur map of $\rho$, denoted $\rho_\alpha^{MP} \boxtimes \rho$, is defined by the Stieltjes transform solving the Marchenko-Pastur equation (Marchenko & Pastur, 1967; Fan & Wang, 2020). It also appears in the free probability literature as the free multiplicative convolution between the probability measures $\rho_{MP}^\alpha$ and $\rho$ (Mingo & Speicher, 2017). When considering the specific case where $\mathbf{\Psi} = \mathbf{I}_P$ (identity matrix of size $P$), then $\rho$ represents the Dirac measure at 1 and we recover the well-known Marchenko-Pastur distribution, denoted as $\rho_{MP}^\alpha$. Furthermore, we denote as $\rho_{MP} \boxtimes^\ell \rho := \rho_{MP} \boxtimes (...(\rho_{MP} \boxtimes \rho))$ the composition of $\ell$ successive Marchenko-Pastur maps.

## 3 BNNs as Modified GP Regression

First we state our definitions of linear-width and sublinear-width regimes.

**Assumption 3.1** (Linear-width regime). *Assume that $\frac{P}{N} \to \alpha$ and $\frac{P}{N_0} \to \alpha_0$ as $P, N, N_0 \to \infty$ with the ratios $\alpha, \alpha_0 \in (0, +\infty)$.*

**Assumption 3.2** (Sublinear-width regime). *Assume that $\frac{P}{N \cdot N_0} \to \gamma$ as $P, N, N_0 \to \infty$ with the ratio $\gamma \in (0, +\infty)$.*

Our first aim in this section is to showcase the emergence of a modified NNGP kernel during the training of BNNs in the linear-width and sublinear-width limits. We then study the Mercer's spectrum of the modified NNGP kernel and exploit it to extend the renormalisation theory to encompass

nonlinear networks in the linear-width regime. Finally, we outline the fundamental arguments supporting the expansion of this theory to the sublinear-width regime.

## 3.1 THE MODIFIED NNGP KERNEL

Mercer's theorem applied to the random kernel $K_{\Theta}^{N,N_0}$ and the data distribution $p_{\mathbf{X}}$ decomposes the kernel into terms of eigenvalues and eigenfunctions as follows: $K_{\Theta}^{N,N_0}(\mathbf{x},\mathbf{x}') = \sum_k \lambda_k^{P,N,N_0} \varphi_k^{P,N,N_0}(\mathbf{x}) \varphi_k^{P,N,N_0}(\mathbf{x}')$. This defines a random spectral measure $\rho_{\Theta}^{P,N,N_0}$ with spectrum given by the eigenvalues and random eigenfunctions $\varphi_k^{P,N,N_0}(\cdot)$ Here, the dependency in $P$ stems from the empirical training data distribution. We use the correspondence between Mercer's eigenvalues and the limiting spectral measure of the corresponding empirical kernel matrix to show that, to estimate the infinite random matrix $\mathbf{K}_{\Theta}^{P,N,N_0}(\mathbf{X},\mathbf{X})$, there is no need to examine the joint distribution of its eigenvalues, as Mercer's eigenvalues can be sampled independently. This is a crucial observation because the correlations between kernel matrix eigenvalues in the classical eigendecomposition is an obstacle in the computation of the posterior distributions.

**Theorem 3.3.** *Assume that Assumption 3.1 (respectively, Assumption 3.2) holds. Assume that for each $k \in \mathbb{N}$ there is a random function $\varphi_k^{\alpha,\alpha_0}: \mathbb{R}^{\mathbb{N}} \mapsto \mathbb{R}$ (respectively, $\varphi_k^{\gamma}$) such that $\varphi_k^{P,N,N_0}(\mathbf{x}_i)$ converges in distribution to $\varphi_k^{\alpha,\alpha_0}(\mathbf{x})$ (respectively, $\varphi_k^{\gamma}(\mathbf{x})$), where $\mathbf{x} \sim \lim_{N_0 \to \infty} \mathbb{P}_{N_0}$. Assume that the spectrum of $\mathbf{K}_{\Theta}^{P,N,N_0}(\mathbf{X},\mathbf{X})$ (respectively, the strictly positive support of the spectrum) admits a limiting nonrandom measure $\rho^{\alpha,\alpha_0}$ (respectively, $\rho^{\gamma}$). Consider the random matrix $\mathbf{\Phi}\mathbf{\Lambda}\mathbf{\Phi}^T$, with $\mathbf{\Phi} \in \mathbb{R}^{P \times M}$, $\mathbf{\Phi}_{i,k} := \varphi_k^{\alpha,\alpha_0}(\tilde{\mathbf{x}}_i)$ (respectively, $\varphi_k^{\gamma}$) and $\mathbf{\Lambda} \in \mathbb{R}^{M \times M}$, $\mathbf{\Lambda}_{k,l} := \delta_{k,l}\lambda_k$ with each $\lambda_k$ follows independently $\rho^{\alpha,\alpha_0}$ (respectively, $\rho^{\gamma}$) and each $\tilde{\mathbf{x}}_i$ follows independently $\lim_{N_0 \to \infty} \mathbb{P}_{N_0}$ [1]. Then, the random matrices $\mathbf{K}_{\Theta}^{P,N,N_0}(\mathbf{X},\mathbf{X})$ and $\mathbf{\Phi}\mathbf{\Lambda}\mathbf{\Phi}^T$ converge (in distribution) to the same distribution over $\mathbb{R}^{\mathbb{N} \times \mathbb{N}}$ in the limit $\frac{M}{P} \to \infty$.*

*Proof.* In the linear-width (respectively, the sub-linear width limit), the positive semi-definiteness of any matrix extracted from $K_{\Theta}^{N,N_0}$ and $p_{\mathbf{X}}$ is maintained (the limit of a positive sequence remains positive), and this suffices to characterise the kernel property over a compact subset of an infinite-dimensional space (Saitoh & Sawano, 2016). Thus, there is a random kernel $K_{\Theta}^{\alpha,\alpha_0}$ (respectively, $K_{\Theta}^{\gamma}$) defined over $\mathbb{R}^{\mathbb{N}}$ which characterises the convergence in distribution of $\mathbf{K}_{\Theta}^{P,N,N_0}(\mathbf{X},\mathbf{X})$. As per Mercer's theorem, $K_{\Theta}^{\alpha,\alpha_0}$ (respectively, $K_{\Theta}^{\gamma}$) also defines a random spectral measure $\rho_{\Theta}^{\alpha,\alpha_0}$ (respectively, $\rho_{\Theta}^{\gamma}$) associated with its Mercer's eigenvalues. By Baker's result (Baker, 1977) stating the convergence of eigenvalues in a kernel matrix to the Mercer eigenvalues of the respective kernel, it follows that $\rho_{\Theta}^{\alpha,\alpha_0}$ (respectively, $\rho_{\Theta}^{\gamma}$) is the limiting spectral distribution of the random matrices $\mathbf{K}_{\Theta}^{P,N,N_0}(\mathbf{X},\mathbf{X})$ in the linear-width limit (respectively, the sublinear-width regime). By assumption, this spectral measure (respectively, the strictly positive support of this spectral measure) is nonrandom $\rho_{\Theta}^{\alpha,\alpha_0} = \rho^{\alpha,\alpha_0}$ (respectively, $\rho_{\Theta}^{\gamma} = \rho^{\gamma}$). Thus, we can reformulate the empirical kernel matrix corresponding to the random kernel $K_{\Theta}^{\alpha,\alpha_0}$ (respectively, $K_{\Theta}^{\gamma}$) as $\mathbf{\Phi}\mathbf{\Lambda}\mathbf{\Phi}^T$, where $\lambda_k$ are drawn independently according to $\rho^{\alpha,\alpha_0}$ (respectively, $\rho^{\gamma}$). Since the spectral measure no longer depends on $\Theta$, the eigenvalues can be sampled independently from the eigenfunctions. It follows that $\mathbf{K}_{\Theta}^{P,N,N_0}(\mathbf{X},\mathbf{X})$ and $\mathbf{\Phi}\mathbf{\Lambda}\mathbf{\Phi}^T$ converge to the same distribution over $\mathbb{R}^{\mathbb{N} \times \mathbb{N}}$. $\square$

This result is non-trivial and only holds if the spectrum admits a *nonrandom* limit: this is the key argument that allows us to disregard, in the limit, the correlations between eigenvalues when using the Mercer decomposition. Note that the distribution of eigenfunctions $\varphi_k^{\alpha,\alpha_0}$ and $\varphi_k^{\gamma}$ are not known in general. We will justify in the next section the SUA as a means for alleviating this limitation. Similarly, we will denote with $\mathbf{\Phi}^*$ evaluations of the eigenfunctions on an unseen data point $\mathbf{x}^*$.

The *modified NNGP kernel* is the random kernel $K_{\Theta}^{\alpha,\alpha_0}$ (respectively, $K_{\Theta}^{\gamma}$) defined over $\mathbb{R}^{\mathbb{N}}$ in the linear-width regime (respectively, the sublinear-width regime). In the limit, the feature map is not known explicitly, but it must exist by Riesz's representation theorem.

---

[1]Expression $\mathbf{\Phi}\mathbf{\Lambda}\mathbf{\Phi}^T$ is not the usual eigendecomposition of a square matrix: the evaluations of eigenfunctions yield rectangular (infinite) matrices. This decomposition is enabled by Mercer's theorem and applies to kernels.

**The nonrandom spectral measure is known in the linear-width regime.** Observe that, in the linear-width regime, for many cases of interest (including ReLU activations), the limiting spectral measure $\rho_\Theta^{\alpha,\alpha_0}$ indeed no longer depends on $\Theta$ and hence becomes a nonrandom measure. To this end, let us first consider the kernel random matrix $\mathbf{K}_{\mathrm{NNGP}}(\mathbf{X},\mathbf{X})$ associated with the NNGP kernel $K_{\mathrm{NNGP}}$. El Harzli et al. (2024) have shown that, under mild assumptions on the activation functions $\sigma^\ell$ (namely measurability and Lipschitz continuity), $\mathbf{K}_{\mathrm{NNGP}}(\mathbf{X},\mathbf{X})$ admits a limiting nonrandom spectral measure $\rho_{\mathrm{NNGP}}^{\alpha_0}$ as $P, N_0 \to \infty$ with constant ratio $\alpha_0$; and furthermore, in the linear-width limit, the limiting spectral distribution of the same random matrix as $\mathbf{K}_\Theta^{P,N,N_0}(\mathbf{X},\mathbf{X})$ but where the interior widths have already been taken to infinity (i.e. when the linear-width limit only pertains to the last-layer width) is $\rho_{MP}^\alpha \boxtimes \rho_{\mathrm{NNGP}}^{\alpha_0}$. By immediate induction, successively applying the linear-width limit to the hidden-layer widths and keeping the remaining interior widths infinite until reaching the input layer, it follows as a direct corollary of Theorem 2 in El Harzli et al. (2024) that, in the linear-width limit, $\mathbf{K}_\Theta^{P,N,N_0}(\mathbf{X},\mathbf{X})$ also admits a limiting nonrandom spectral measure given by the composed Marchenko-Pastur maps $\rho_{MP}^\alpha \boxtimes^L \rho_{\mathrm{NNGP}}^{\alpha_0}$.[2]

## 3.2 Training BNNs with the Modified NNGP Kernel

We can now study the predictor statistics of trained BNNs in the linear-width limit and the sublinear-width limit. In particular, the following theorem provides integral formulae to estimate, under the SUA, the first and second moments of the trained BNN using only the limiting spectral measure. In this section, the results hold indistinctly of the linear-width or the sublinear-width limit, so to simplify notations, we will note $\rho$ for both $\rho^{\alpha,\alpha_0}$ and $\rho^\gamma$ and $K_\Theta$ for both $K_\Theta^{\alpha,\alpha_0}$ and $K_\Theta^\gamma$.

**Theorem 3.4.** *Assume that Assumption 3.1 or Assumption 3.2 holds. Let $\rho$ be the nonrandom spectral measure characterising the modified NNGP kernel $K_\Theta$, and assume that the SUA holds.*

*The mean $\langle f \rangle(\mathbf{x}^*, \mathbf{X}, \mathbf{y})$ and variance $\langle (\delta f)^2 \rangle(\mathbf{x}^*, \mathbf{X}, \mathbf{y})$ of the predictor associated to a BNN with training data $(\mathbf{X}, \mathbf{y})$ is given by expressions equation 2 and equation 3 respectively:[3]*

$$\langle f \rangle = \int \left( \mathbf{\Phi}^{*T} \mathbf{\Lambda} \mathbf{\Phi}^T \mathbf{\Phi}^{T\dagger} \mathbf{\Lambda}^{-1} \mathbf{\Phi}^\dagger \mathbf{y} \right) \cdot \frac{p(\mathbf{y}, \mathbf{\Phi} | \mathbf{\Lambda}, \mathbf{X})}{p(\mathbf{y}|\mathbf{X})} \mathrm{d}\rho\left(\mathbf{\Lambda}\right) \mathcal{D}\mathbf{\Phi}\mathcal{D}\mathbf{\Phi}^* \tag{2}$$

$$\langle (\delta f)^2 \rangle = \int (\mathbf{\Phi}^{*T} \mathbf{\Lambda} \mathbf{\Phi}^* - \mathbf{\Phi}^{*T} \mathbf{\Lambda} \mathbf{\Phi}^T \mathbf{\Phi}^{T\dagger} \mathbf{\Lambda}^{-1} \mathbf{\Phi}^\dagger \mathbf{\Phi} \mathbf{\Lambda} \mathbf{\Phi}^*) \cdot \frac{p(\mathbf{y}, \mathbf{\Phi} | \mathbf{\Lambda}, \mathbf{X})}{p(\mathbf{y}|\mathbf{X})} \mathrm{d}\rho\left(\mathbf{\Lambda}\right) \mathcal{D}\mathbf{\Phi}\mathcal{D}\mathbf{\Phi}^* \tag{3}$$

*where $\mathcal{D}\mathbf{\Phi}$ is a standard Gaussian matrix measure, $\mathbf{\Phi}_{i,j} \sim_{\mathrm{iid}} \mathcal{N}(\mu_{K_\Theta}, \sigma_{K_\Theta}^2)$ obtained from the SUA; the likelihood is given by $p(\mathbf{y}, \mathbf{\Phi}|\mathbf{\Lambda}, \mathbf{X}) \sim \mathcal{N}(\mathbf{\Phi}^T \mathbf{y}, \mathbf{\Lambda})$; the marginal likelihood is given by $p(\mathbf{y}|\mathbf{X}) = \int p(\mathbf{y}, \mathbf{\Phi}|\mathbf{\Lambda}, \mathbf{X})\mathrm{d}\rho\left(\mathbf{\Lambda}\right) \mathcal{D}\mathbf{\Phi}$.*

The integral forms equation 2 and equation 3 provide a new estimation of the predictor statistics of a trained BNN. While these expressions are exact only in the limit, we will present empirical evidence suggesting that they constitute a reasonable approximation. A practical challenge arises from the need to estimate the spectral distribution $\rho(\mathbf{\Lambda})$, which often involves diagonalising a kernel matrix. This process can be computationally intensive, especially for large training datasets; however, it is worth mentioning that in many applications of BNNs, where the training datasets are relatively small, this computational difficulty becomes less significant. In particular, in the linear-width regime, $\rho(\mathbf{\Lambda})$ is the Marchenko-Pastur map of the empirical spectral distribution of the NNGP kernel $\rho_{MP}^\alpha \boxtimes^L \rho_{\mathrm{NNGP}}^{\alpha_0}$ which can be computed by diagonalising the NNGP kernel Cho & Saul (2009) to estimate $\rho_{\mathrm{NNGP}}^{\alpha_0}$ and solving numerically the Marchenko-Pastur fixed-point equation in the Stieltjes transform space (Marchenko & Pastur, 1967; Fan & Wang, 2020).

Theorem 3.4 and its proof also offer valuable new perspectives on the applicability of the SUA. In particular, in the last steps of the proof, the probability density of $\mathbf{\Phi}, \mathbf{\Phi}^*$ no longer appears directly in the integral. For given $\mathbf{\Lambda}, \mathbf{X}, \mathbf{x}^*$, if each orthogonal matrix $\mathbf{\Phi}, \mathbf{\Phi}^*$ has a non-zero probability of

---

[2]This result first appeared in the context of neural networks in Fan & Wang (2020). The result by El Harzli et al. (2024) extends it to a more general setting.

[3]Here, $\mathbf{X}$ is the infinite matrix representing the linear-width or sublinear-width limits of the training data. To be completely rigorous, we should write $f(\mathbf{x}^*, \mathbf{X}, \mathbf{y}) = \lim_{P,N,N_0 \to \infty} f_{P,N,N_0}(\mathbf{x}^*{}_{N_0}, \mathbf{X}_{P,N_0}, \mathbf{y}_P)$, but by slight abuse of notation the same notation is used for both. In practice, one would use finite (but large) objects in calculations.

occurring, the integral spans uniformly over the entire space of orthogonal matrices of size $P \times M$. This is useful because in the limit of infinite dimensions, this space coincides with that of Gaussian matrices with independent entries (independent infinite Gaussian vectors are orthogonal). Remarkably, this property precisely corresponds to the SUA in kernel theory (Karoui, 2010; Jacot et al., 2020; Simon et al., 2023), which posits that, in terms of the generalisation error statistics in kernel regression, the eigenfunctions can be approximated by Gaussian matrices with independent entries, denoted $\Phi_{i,k} \sim \mathcal{N}(\mu_K, \sigma_K^2)$. Note that this approximation is not the same as the Gaussian equivalence assumption (Schröder et al., 2023; Cui et al., 2023), which assumes Gaussianity of the predictor (here, the eigenfunctions are Gaussian but the predictor is non-Gaussian). To the best of our knowledge, this marks a first connection between BNNs and the SUA from kernel theory (i.e. applied to Mercer's eigenfunctions).

We can now reframe the question concerning the correctness of the SUA approximation as follows: given $\Lambda$ and $\mathbf{X}$, is it the case that all orthogonal matrices $\Phi$ have non-zero probabilities (according to $\Theta$) to satisfy $\mathbf{K}_\Theta(\mathbf{X}, \mathbf{X}) = \Phi\Lambda\Phi^T$? If this condition holds, the SUA is applicable and Gaussian eigenfunctions can be used for estimating equation 2 and equation 3. Since the prior is a Gaussian matrix, any matrix has a non-zero probability of occurrence, thus it suffices to show that for any orthogonal $\Phi$, there exists a $\Theta$ such that $\mathbf{K}_\Theta(\mathbf{X}, \mathbf{X}) = \Phi\Lambda\Phi^T$. In particular, it is easy to show that the SUA always holds in the linear case: for any orthogonal $\Phi$, there exists $\Theta$ such that $\mathbf{X}^T\Theta^T\Theta\mathbf{X} = \Phi\Lambda\Phi^T$. With a non-linearity, the problem is less obvious: is there a $\Theta$ such that $\phi(\Theta, \mathbf{X})^T\phi(\Theta, \mathbf{X}) = \Phi\Lambda\Phi^T$ for any orthogonal $\Phi$ ? In the next section, we show that, in the linear-width limit, this criterion about $\phi(\Theta, .)$ and $\mathbf{X}$ is a necessary and sufficient assumption for the renormalisation theory to hold.

### 3.3 AN EXTENDED RENORMALISATION THEORY

This section only concerns the linear-width regime. In this limit, we can explicitly derive the results of our integral estimators because the corresponding limiting nonrandom spectral measure is known El Harzli et al. (2024) (see Paragraph 3.1).

The renormalisation theory for linear BNNs establishes that, in the linear-width limit, the marginal likelihood $p(\mathbf{y}|\mathbf{X})$ follows a multivariate Gaussian with mean vector $\mathbf{y}$ and covariance matrix $u_0^L\mathbf{K}_0$, with $\mathbf{K}_0 = \frac{1}{N_0}\mathbf{X}\mathbf{X}^T$ and $u_0$ the renormalisation factor fulfilling the fixed-point equation $1 - u_0 = \alpha(1 - \frac{r_0}{u_0^L})$ with $r_0 = \frac{1}{P}\mathbf{y}^T\mathbf{K}_0^{-1}\mathbf{y}$. This result was obtained in Li & Sompolinsky (2021) by successively applying the saddle point method when integrating out the weights $\Theta, \mathbf{W}^L$.

The following theorem shows that this result generalises to BNNs with nonlinear activations if and only if the SUA is correct (i.e., it gives the correct estimate for the marginal likelihood). Here, we exploit the characterisation of the correctness of the SUA developed as a corollary of Theorem 3.4.

**Theorem 3.5.** *Assume that Assumption 3.1 holds. Let $u_{\text{NNGP}}$ fulfil the fixed-point equation*

$$1 - u_{\text{NNGP}} = \alpha(1 - \frac{r_{\text{NNGP}}}{u_{\text{NNGP}}^L}) \tag{4}$$

*with $r_{\text{NNGP}} = \frac{1}{P}\mathbf{y}^T\mathbf{K}_{\text{NNGP}}^{-1}\mathbf{y}$. The marginal likelihood for a nonlinear BNN verifies $p(\mathbf{y}|\mathbf{X}) \sim \mathcal{N}(\mathbf{y}, u_{\text{NNGP}}^L\mathbf{K}_{\text{NNGP}})$ if and only if, for given $\Lambda, \mathbf{X}$ and orthogonal $\Phi$, there exists $\Theta$ such that $\phi(\Theta, \mathbf{X})^T\phi(\Theta, \mathbf{X}) = \Phi\Lambda\Phi^T$.*

*Proof.* In the linear case, the true NNGP kernel is simply $K_{\text{NNGP}}(\mathbf{x}, \mathbf{x}') = \frac{1}{N_0}\mathbf{x}^T\mathbf{x}'$; hence, $\rho_{\text{NNGP}}^{\alpha_0}$ is the limiting spectral distribution of the kernel random matrix $\mathbf{K}_0$, which we denote $\rho_0$. The renormalisation theory of linear networks thus implies that:

$$\int p(\mathbf{y}, \Phi|\Lambda, \mathbf{X})\mathrm{d}\left(\rho_{MP}^\alpha \boxtimes^L \rho_0\right)(\Lambda)\mathcal{D}\Phi \sim \mathcal{N}(\mathbf{y}, u_0^L\mathbf{K}_0) \tag{5}$$

This identity is exact in the linear-width limit and holds in general without assumption on $\mathbf{X}, \mathbf{y}$, as long as the integral $\mathcal{D}\Phi$ is uniform on the space of orthogonal matrices.

Assume the SUA holds in the nonlinear case. Thus, we can express the marginal likelihood as $p(\mathbf{y}|\mathbf{X}) = \int p(\mathbf{y}, \Phi|\Lambda, \mathbf{X})\mathrm{d}\left(\rho_{MP}^\alpha \boxtimes^L \rho_{\text{NNGP}}^{\alpha_0}\right)(\Lambda)\mathcal{D}\Phi$. Furthermore, we can freely interchange

the role of $\mathbf{K}_0$ and $\mathbf{K}_{\mathrm{NNGP}}(\mathbf{X}, \mathbf{X})$ in equation 5. Indeed, it suffices to consider the linear case and a new training dataset $\tilde{\mathbf{X}}$ which exhibits the same covariance structure $\frac{1}{N_0}\tilde{\mathbf{X}}\tilde{\mathbf{X}}^T$ as that of $\mathbf{K}_{\mathrm{NNGP}}(\mathbf{X}, \mathbf{X})$. As a result, a similar equation to equation 5 applies to nonlinear networks by replacing the linear kernel $(\mathbf{x}, \mathbf{x}') \mapsto \frac{1}{N_0}\mathbf{x}^T\mathbf{x}'$ with the true NNGP kernel in the equations, provided that the SUA holds:

$$\int p(\mathbf{y}, \boldsymbol{\Phi}|\boldsymbol{\Lambda}, \mathbf{X})\mathrm{d}\left(\rho_{MP}^{\alpha} \boxtimes^L \rho_{\mathrm{NNGP}}^{\alpha_0}\right)(\boldsymbol{\Lambda})\,\mathcal{D}\boldsymbol{\Phi} \sim \mathcal{N}(\mathbf{y}, u_{\mathrm{NNGP}}^L\mathbf{K}_{\mathrm{NNGP}}) \tag{6}$$

Conversely, if the SUA does not hold, the integral with respect to $\boldsymbol{\Phi}$ does not span the space of orthogonal matrices, the identity equation 6 is no longer exact (all integrands are strictly positive), nor is the renormalisation. Thus, the SUA is necessary and sufficient for the renormalisation to hold. $\qquad\square$

This result characterises the renormalisation theory in the nonlinear case and describes a continuous transition between an accurate and a poor approximation. Specifically, if the SUA significantly deviates (the feature map spans a small fraction of the space of orthogonal matrices) then the equivalence equation 6 also deviates substantially from the correct value. For example, in the spiked kernel case, which occurs for one step of feature learning Dandi et al. (2024), the orthogonal matrices permissible for constructing the prior kernel is significantly constrained, thus we anticipate that the spectral universality assumption would fail in this scenario. Conversely, if the SUA is nearly accurate (meaning that the feature map encompasses a large portion of the space of orthogonal matrices) then equation 6 closely approximates the true marginal likelihood. Thanks to these insights, future research on BNNs can benefit from research advances on the accuracy of the SUA (Liu et al., 2021).

### 3.4 Application to the Sublinear-Width Regime

In this section, we consider the application of our integral estimators to the sublinear-width regime.

Assume that Assumption 3.2 holds. In this regime, the ratios $\alpha = \frac{P}{N}$ and $\alpha_0 = \frac{P}{N_0}$ from the linear-width regime tend to infinity and hence are no longer bounded. Here, the renormalisation theory breaks even in the linear case, because the random matrix $\mathbf{K}_\Theta^{P,N,N_0}(\mathbf{X}, \mathbf{X})$ becomes degenerate and its limiting spectral distribution is the Dirac distribution at 0 and equation 5 no longer holds. A mismatch with the predictions of the renormalisation theory has indeed been observed empirically for high values of $\alpha$ and $\alpha_0$ (Li & Sompolinsky, 2021), hence the need for a new theory.

Remarkably, our kernel-theoretic description of BNNs (Theorem 3.4) still holds as its validity relies only on the dot product of random feature maps $\phi(\Theta, \cdot)$ defining a random kernel (see proof of Theorem 3.3). This remains true for the sublinear-width regime (as well as for other regimes of interest). Additionally, zero eigenvalues in Mercer's decomposition can be disregarded since they do not contribute to the kernel evaluation. An alternative perspective is that, when calculating the first two moments, one takes into account the Moore-Penrose pseudo-inverse $\mathbf{K}_\Theta^{P,N,N_0}(\mathbf{X}, \mathbf{X})^\dagger$ of the kernel random matrix. Consequently, only the contributions from the *strictly positive support* of the limiting spectral distribution are considered (see Theorem 3.3). As a result, our integral estimators for the mean and variance of the predictor remain applicable under the SUA. The only missing element in the argument is whether $p(\boldsymbol{\Lambda}|\mathbf{X}, \mathbf{x}^*)$, which is now the strictly positive support of the limiting spectral distribution of $\mathbf{K}_\Theta^{P,N,N_0}(\mathbf{X}, \mathbf{X})$ (rescaled to integrate to 1), also converges to a nonrandom spectral measure (in order to apply Theorem 3.3). Thus, precisely characterising the asymptotic behavior of this Mercer's random spectral measure and assessing the SUA in this regime is an interesting avenue for further research. Note however that this approach only concerns the predictor statistics and is not derived in weight space, thus one limitation of the approach that we anticipate is that it might be difficult to characterize (strong) feature learning from this standpoint.

Although we don't have an analytical formula for the limiting spectral measure, it is still possible to numerically compute the strictly positive support of the random matrix $\mathbf{K}_\Theta^{P,N,N_0}(\mathbf{X}, \mathbf{X})$ and use our integral forms equation 7 and equation 8 to estimate the predictor of a trained BNNs in this regime.

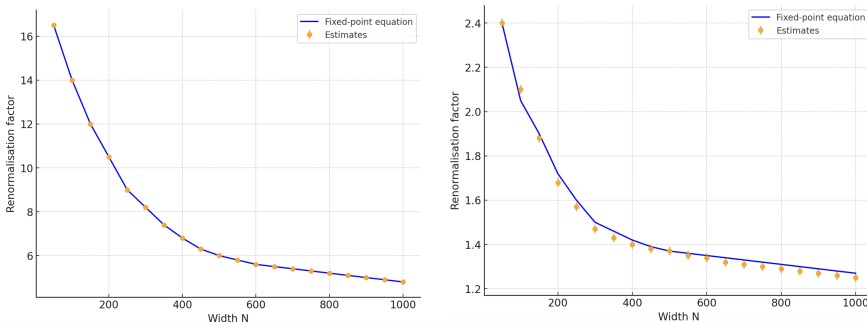

Figure 1: Comparison with Li & Sompolinsky (2021) in the linear-width regime. On the left, the linear setting on our synthetic dataset with $N_0 = 500$ and $P = 200$. On the right, the nonlinear setting on the subset of MNIST. In both cases, the blue line is computed using the fixed-point equation 4, and the orange dots are the ratio between the result of our integral estimator equation 3 and the variance of Bayesian linear regression (respectively, NNGP regression) on the left (respectively, on the right). In the nonlinear case, we use a large width $\hat{N} = 10000$ to estimate the NNGP kernel matrix for ReLU.

## 4 EXPERIMENTS

We consider a synthetic dataset generated by a multivariate Gaussian $\mathbf{x} \sim \mathcal{N}(0, \frac{1}{N_0}\mathbf{I}_{N_0})$ to which we apply a linear teacher and noise $y = \beta^T\mathbf{x} + \epsilon$ with $\epsilon \sim \mathcal{N}(0, \sigma_\epsilon^2)$ and $\beta = (1, ..., 1)^T$. We also consider a subset of MNIST restricted to classes "0" and "1" of size $P = 105$ and with $N_0 = 784$ pixels per image.

Our first experiment verifies that our estimators coincide with the predictions of the renormalisation theory in the linear-width limit both for a single hidden-layer network with ReLU activations and a linear network with a hidden layer. We computed the renormalisation factors using the fixed-point equation 4 and used equation 2 and equation 3 to estimate the mean and the variance of the predictor in our approach. To compute equation 2 and equation 3 we first computed the Marchenko-Pastur maps of the empirical spectral distributions (of the NNGP kernel) by solving numerically the Marchenko-Pastur fixed-point equation in the Stieltjes transform space (Marchenko & Pastur, 1967; Fan & Wang, 2020); then, we relied on the SUA to estimate the integral forms. In a second experiment, we simulated the sublinear-width regime $P \propto N \cdot N_0$ (for which the renormalisation theory breaks, see Figure 14 in Li & Sompolinsky (2021)) using a small value of $N_0$ (thus making $\alpha_0$ high). We compared our estimators for the regime as described in the previous section with the predictions of BNNs trained with variational inference using the library Pyro (Bingham et al., 2019). For the spectral distribution, we computed the strictly positive support of the empirical spectral distributions by sampling and diagonalising the empirical kernel matrices several times and shuffling the eigenvalues; we continued to use the SUA for eigenfunctions.

As shown in Figure 1, our estimates align with the renormalisation theory in the linear-width limit. As shown in Figure 2, for a regime where the renormalisation theory is inaccurate, our estimators provide reasonable matches to the actual predictions. These results suggest that our estimates are better suited to regimes where key assumptions of the renormalisation theory do not hold. Sources of discrepancies include: the fact that we used finite values of $P, N, N_0$ (whereas the theory is only exact in the limit); the SUA may not be fully accurate in this configuration; the limiting spectral measure may not be nonrandom.

## 5 CONCLUSION

In this paper, we have explored bridges between BNNs trained under interesting idealised limits and kernel theory, which enable an extension of the renormalisation theory to non-linear networks. From a practical standpoint, our theory offers a new way to estimate the prediction of BNNs with better accuracy in the sublinear-width regime. Finally, we hope that the theory developed here will

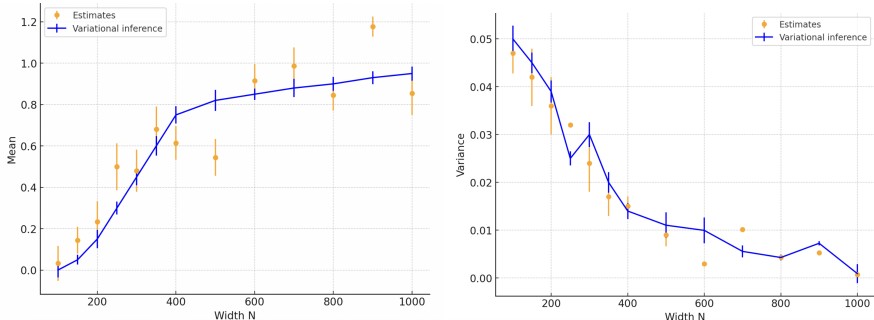

Figure 2: Sublinear-width regime. Mean and variance of the predictor against the width $N$ of the single ReLU hidden-layer on our synthetic dataset with $P = 200$ and $N_0 = 40$. In both cases, the blue line is computed using the probabilistic predictions of a BNN trained with variational inference on the synthetic data, and the orange dots correspond to our integral estimates.

motivate further research on the application of existing kernel-theoretic results in the context of BNNs.

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

# A    PROOF OF THEOREM 3.4

We calculate the conditional expectation $\langle f \rangle(\mathbf{x}^*, \mathbf{X}, \mathbf{y}, \Theta)$ and variance $\langle (\delta f)^2 \rangle(\mathbf{x}^*, \mathbf{X}, \mathbf{y}, \Theta)$ of the predictor by marginalising over the readout weights $\mathbf{W}^L$:

$$\langle f \rangle(\mathbf{x}^*, \mathbf{X}, \mathbf{y}, \Theta) = \int {\mathbf{W}^L}^T \phi(\Theta, \mathbf{X}) p(\mathbf{W}^L | \mathbf{X}, \mathbf{y}, \Theta) \mathrm{d}\mathbf{W}^L$$

$$\langle (\delta f)^2 \rangle(\mathbf{x}^*, \mathbf{X}, \mathbf{y}, \Theta) = \int \left[ {\mathbf{W}^L}^T \phi(\Theta, \mathbf{X}) \right]^2 p(\mathbf{W}^L | \mathbf{X}, \mathbf{y}, \Theta) \mathrm{d}\mathbf{W}^L - [f(\mathbf{x}^*, \mathbf{X}, \mathbf{y}, \Theta)]^2$$

where $p(\mathbf{W}^L | \mathbf{X}, \mathbf{y}, \Theta)$ can be expressed by Bayes rule using Gaussian likelihoods. The result can be expressed analytically and yields the same prediction as GP regression with prior $\mathcal{GP}(0, K_\Theta^{N,N_0})$:

$$\langle f \rangle(\mathbf{x}^*, \mathbf{X}, \mathbf{y}, \Theta) = [\mathbf{k}_\Theta^{P,N,N_0}(\mathbf{x}^*, \mathbf{X})]^T [\mathbf{K}_\Theta^{P,N,N_0}(\mathbf{X}, \mathbf{X})]^{-1} \mathbf{y}$$

$$\langle (\delta f)^2 \rangle(\mathbf{x}^*, \mathbf{X}, \mathbf{y}, \Theta) = K_\Theta^{P,N,N_0}(\mathbf{x}^*, \mathbf{x}^*) - [\mathbf{k}_\Theta^{P,N,N_0}(\mathbf{x}^*, \mathbf{X})]^T [\mathbf{K}_\Theta^{P,N,N_0}(\mathbf{X}, \mathbf{X})]^{-1} \mathbf{k}_\Theta^{P,N,N_0}(\mathbf{x}^*, \mathbf{X}).$$

To marginalise over $\Theta \sim p(\Theta | \mathbf{X}, \mathbf{y})$, we perform the change of variables $\Theta \mapsto (\mathbf{\Phi}^*, \mathbf{\Phi}, \mathbf{\Lambda})$, relying on the fact that all quantities of interest involving $\Theta$ can be expressed in the limit solely using eigenvalues and eigenfunctions, namely $\mathbf{K}_\Theta(\mathbf{X}, \mathbf{X}) = \mathbf{\Phi}\mathbf{\Lambda}\mathbf{\Phi}^T$, $\mathbf{k}_\Theta(\mathbf{x}^*, \mathbf{X}) = \mathbf{\Phi}\mathbf{\Lambda}\mathbf{\Phi}^*$, and $K_\Theta(\mathbf{x}^*, \mathbf{x}^*) = \mathbf{\Phi}^{*T}\mathbf{\Lambda}\mathbf{\Phi}^*$. Since $\mathbf{\Phi} \in \mathbb{R}^{P \times M}$ has orthogonal rows, $\mathbf{\Phi}^\dagger = \mathbf{\Phi}^T \left( \mathbf{\Phi}\mathbf{\Phi}^T \right)^{-1}$, and $\mathbf{\Phi}^{T\dagger} = \left( \mathbf{\Phi}\mathbf{\Phi}^T \right)^{-1} \mathbf{\Phi}$. This allows us to express the mean and variance of the predictor as follows:

$$\langle f \rangle = \int \left( \mathbf{\Phi}^{*T}\mathbf{\Lambda}\mathbf{\Phi}^T \mathbf{\Phi}^{T\dagger}\mathbf{\Lambda}^{-1}\mathbf{\Phi}^\dagger \mathbf{y} \right) \cdot p(\mathbf{\Lambda}, \mathbf{\Phi}, \mathbf{\Phi}^* | \mathbf{X}, \mathbf{x}^*) \cdot \frac{p(\mathbf{y} | \mathbf{\Lambda}, \mathbf{\Phi}, \mathbf{\Phi}^*, \mathbf{X}, \mathbf{x}^*)}{p(\mathbf{y} | \mathbf{X})} \mathrm{d}\mathbf{\Lambda}\mathrm{d}\mathbf{\Phi}\mathrm{d}\mathbf{\Phi}^*$$
$$(7)$$

$$\langle (\delta f)^2 \rangle = \int \left( \mathbf{\Phi}^{*T}\mathbf{\Lambda}\mathbf{\Phi}^* - \mathbf{\Phi}^{*T}\mathbf{\Lambda}\mathbf{\Phi}^T \mathbf{\Phi}^{T\dagger}\mathbf{\Lambda}^{-1}\mathbf{\Phi}^\dagger \mathbf{\Phi}\mathbf{\Lambda}\mathbf{\Phi}^* \right) \cdot$$
$$p(\mathbf{\Lambda}, \mathbf{\Phi}, \mathbf{\Phi}^* | \mathbf{X}, \mathbf{x}^*) \frac{p(\mathbf{y} | \mathbf{\Lambda}, \mathbf{\Phi}, \mathbf{\Phi}^*, \mathbf{X}, \mathbf{x}^*)}{p(\mathbf{y} | \mathbf{X})} \mathrm{d}\mathbf{\Lambda}\mathrm{d}\mathbf{\Phi}\mathrm{d}\mathbf{\Phi}^* \quad (8)$$

Furthermore, it holds that $p(\mathbf{\Lambda}, \mathbf{\Phi}, \mathbf{\Phi}^* | \mathbf{X}, \mathbf{x}^*) = p(\mathbf{\Lambda} | \mathbf{X}, \mathbf{x}*)p(\mathbf{\Phi}, \mathbf{\Phi}^* | \mathbf{\Lambda}, \mathbf{X}, \mathbf{x}^*)$ if $p(\mathbf{\Lambda} | \mathbf{X}, \mathbf{x}^*) \neq 0$ and also $p(\mathbf{y} | \mathbf{\Lambda}, \mathbf{\Phi}, \mathbf{\Phi}^*, \mathbf{X}, \mathbf{x}^*) = \frac{p(\mathbf{y}, \mathbf{\Phi}, \mathbf{\Phi}^* | \mathbf{\Lambda}, \mathbf{X}, \mathbf{x}^*)}{p(\mathbf{\Phi}, \mathbf{\Phi}^* | \mathbf{\Lambda}, \mathbf{X}, \mathbf{x}^*)}$ if $p(\mathbf{\Phi}, \mathbf{\Phi}^* | \mathbf{\Lambda}, \mathbf{X}, \mathbf{x}^*) \neq 0$, which yields:

$$\langle f \rangle = \int \left( \mathbf{\Phi}^{*T}\mathbf{\Lambda}\mathbf{\Phi}^T \mathbf{\Phi}^{T\dagger}\mathbf{\Lambda}^{-1}\mathbf{\Phi}^\dagger \mathbf{y} \right) \cdot p(\mathbf{\Lambda} | \mathbf{X}, \mathbf{x}^*) \frac{p(\mathbf{y}, \mathbf{\Phi}, \mathbf{\Phi}^* | \mathbf{\Lambda}, \mathbf{X}, \mathbf{x}^*)}{p(\mathbf{y} | \mathbf{X})} \mathrm{d}\mathbf{\Lambda}\mathrm{d}\mathbf{\Phi}\mathrm{d}\mathbf{\Phi}^*$$

$$\langle (\delta f)^2 \rangle = \int \left( \mathbf{\Phi}^{*T}\mathbf{\Lambda}\mathbf{\Phi}^* - \mathbf{\Phi}^{*T}\mathbf{\Lambda}\mathbf{\Phi}^T \mathbf{\Phi}^{T\dagger}\mathbf{\Lambda}^{-1}\mathbf{\Phi}^\dagger \mathbf{\Phi}\mathbf{\Lambda}\mathbf{\Phi}^* \right) \cdot$$
$$p(\mathbf{\Lambda} | \mathbf{X}, \mathbf{x}^*) \frac{p(\mathbf{y}, \mathbf{\Phi}, \mathbf{\Phi}^* | \mathbf{\Lambda}, \mathbf{X}, \mathbf{x}^*)}{p(\mathbf{y} | \mathbf{X})} \mathrm{d}\mathbf{\Lambda}\mathrm{d}\mathbf{\Phi}\mathrm{d}\mathbf{\Phi}^*$$

where the integral over $\mathbf{\Lambda}$ is restricted to segments where $p(\mathbf{\Lambda} | X, x^*) \neq 0$ and the integrals over $\mathbf{\Phi}$ and $\mathbf{\Phi}^*$ are restricted to where $p(\mathbf{\Phi}, \mathbf{\Phi}^* | \mathbf{\Lambda}, X, x^*) \neq 0$. We obtain equations equation 2 and equation 3 by replacing $\mathrm{d}\mathbf{\Phi}$ and $\mathrm{d}\mathbf{\Phi}^*$ by standard Gaussian matrix measures and the density of $\mathbf{\Lambda}$ by the spectral measure.

