# OpenReview forum: "Bayesian Treatment of the Spectrum of the Empirical Kernel in (Sub)Linear-Width Neural Networks"
_ICLR.cc/2025/Conference — ICLR 2025 Poster_

### Official Review · Reviewer_H2W1 · 2024-10-28

**Soundness:** 2
**Presentation:** 2
**Contribution:** 3
**Rating:** 6
**Confidence:** 4

**Summary:**

The authors propose combining ideas from random matrix theory to evaluate limits of infinite width Bayesian neural networks (such as the classic work by Neal (1996)) when the number of observations also goes to infinity as well as the number of input dimensions. The paper provides novel claims of the behavior of the limit in sublinear regimes.

**Strengths:**

Using Marchenko-Pastur to evaluate this type of limits, seems like a good idea. Combining these approaches with Mercer's theorem seems valuable. The ideas of random matrix theory in general seem underutilized in the ML community.

**Weaknesses:**

1. It is not immediately clear how the integrals in Theorem 3.4 can be explicitly evaluated.
2. Notation for $\Phi^*$ in Theorem 3.4 was never defined.
3. Between lines 300 and 303, the authors claim that for __small__ datasets the computations would not be too computationally intensive. However, in footnote 3 (p. 6), they claim that the limit can be approximated using __large__ objects. It is not completely clear what has to be large or small for the evaluations to be feasible.
4. Section 4 appears to be incomplete. It is not evident from the writing which of the figures correspond to the two examples they mention at the beginning of the section.
5. In the first experiment they mention, with the "linear teacher", the authors do not specify how the $\beta$ is defined.

**Questions:**

1. For Figures 1 and 2, how many simulations are the authors using to obtain the error bars? Also, is there any uncertainty around the blue lines in the same figures? If so, it should be reported.
2. Can the authors reconcile the differences between the simulations in Figure 2 between their proposed estimates and the blue lines?
3. In Theorem 3.3, the authors bring in the notation:
$\mathbf{x}\sim \lim\_{{N_0}\to\infty} \mathbb{P}\_{N_0},$ without having defined $\mathbb{P}\_{N_0}$.
Does $\mathbb{P}\_{N_0}$ correspond to $p\_\mathbf{x}$ for a specific dimension of $N\_0$?
4. The authors emphasize the rectified linear unit (ReLU) activation function (deservedly so). However I wonder if this approach can also work for other activation functions?

---

> ### Author Response · Authors · 2024-11-19
>
> Comment: It is not immediately clear how the integrals in Theorem 3.4 can be explicitly evaluated.
>
> Answer: In the integrals in Theorem 3.4, the integrands are explicitly known as functions of the integration variables $\Lambda, \Phi, \Phi^*$ and the measures over the integration variables are also known: Marchenko-Pastur map for $\Lambda$ and isotropic multivariate Gaussian for $\Phi, \Phi^*$. Given this, it suffices to sample $\Lambda, \Phi, \Phi^*$ according to these measures and to compute the integrands to calculate the integrals. The least straightforward step is to compute the Marcheko-Pastur map and we gave more details about this step in l. 424-426 in our initial submission (l. 431-472 in our revised submission).
>
> Comment: Notation for $\Phi^*$ in Theorem 3.4 was never defined.
>
> Answer: The notation was defined in l. 255 in our initial submission (l. 259 in our revised submission).
>
> Comment: Between lines 300 and 303, the authors claim that for small datasets the computations would not be too computationally intensive. However, in footnote 3 (p. 6), they claim that the limit can be approximated using large objects. It is not completely clear what has to be large or small for the evaluations to be feasible.
>
> Answer: The footnote highlights that our results are theoretically established in the limit of large dimensions. However, because of the kernel matrix inversion, dealing with large-scale datasets in practice is challenging. To rest the theory, we must therefore constrain ourselves to a setting where the dimensions are large enough for the quantities to have converged to their theoretical limits but small enough for the computations to be feasible. This is easier to achieve for simple, synthetic, datasets where the convergence to theoretical limits is quick, and more difficult for more complex real-world datasets. This partly explains the discrepancies we observe between our theory and experiments. However, please note that this is a shortcoming that the entire community working on theoretical infinite limits (especially with kernels) has to deal with, not a particular weakness of our work.
>
>
> Question: Section 4 appears to be incomplete. It is not evident from the writing which of the figures correspond to the two examples they mention at the beginning of the section.
>
> Answer: We have refined the presentation of our experiments to improve readability. Please see Figures 1 and 2 in our revised submission.
>
> Question: In the first experiment they mention, with the "linear teacher", the authors do not specify how the $\beta$ is defined.
>
> Answer: We have added the value we took for $\beta$ l. 42 in our revised submission.
>
> Question: For Figures 1 and 2, how many simulations are the authors using to obtain the error bars? Also, is there any uncertainty around the blue lines in the same figures? If so, it should be reported.
>
> Answer: We are using 10 simulations to obtain the error bars. Please see the improvements to our Figures in our revised submission.
>
> Question: Can the authors reconcile the differences between the simulations in Figure 2 between their proposed estimates and the blue lines?
>
> Answer: We have listed the sources of discrepancies in l. 476-479 in our initial submission (l. 483-485 in our revised submission): "the fact that we used finite values of P, N, N0 (whereas the theory is only exact in the limit); the SUA may not be fully accurate in this configuration; the limiting spectral measure may not be nonrandom."
>
> Question: In Theorem 3.3, the authors bring in the notation $\lim_{\infty} P_{N_0}$ without having defined [...]. Does $P_{N_0}$
>  correspond to $p_{X}$ for a specific dimension?
>
>  Answer: $P_{N_0}$ (and $\lim_{\infty} P_{N_0}$) was defined in the preliminaries in l. 131-133 in our initial submission (l. 133-135 in our revised submission).
>
>  Question: The authors emphasize the rectified linear unit (ReLU) activation function (deservedly so). However I wonder if this approach can also work for other activation functions?
>
>  Answer: Yes, our approach does work for other activation functions, as stated in l. 260-261 in our initial submission for instance (l. 266-272 in our revised submission): the result from El Harzli et al. 2024 is established under very general assumptions on the activation function, namely measurability and Lipschitz continuity.

---

> > ### Comment · Reviewer_H2W1 · 2024-11-20
> >
> > Thank you for answering my questions and pointing out the specific lines I had missed. I have revised my rating accordingly.
> >
> > I would recommend moving the indications on how the computations are done (i.e., 431-472) to Section 3 rather than having it in Section 4. Having theoretical/methodological developments in the Experiments section seems a tad disorganized.

---

> > > ### Author Response · Authors · 2024-11-21
> > >
> > > Thank you for your suggestion. We have added the details of numerical computations to the section 3 (see l.308-312 in our revised submission).

---

### Official Review · Reviewer_U5Zu · 2024-10-29

**Soundness:** 3
**Presentation:** 3
**Contribution:** 3
**Rating:** 8
**Confidence:** 2

**Summary:**

The paper gives integral formulas describing the outputs of BNNs in the linear and sublinear width regimes.

**Strengths:**

The theoretical contributions are extremely strong, especially around the integral formulas describing the outputs of BNNs in the linear and sublinear width regimes.

**Weaknesses:**

The primary weakness is around the experimental results.  These are restricted to MLPs on very simple datasets.  Though I would be happy to consider an argument that the results should generalise and/or that it would be prohibitively difficult to get results outside this setting.

The experimental results are presented very poorly.  For instance:
* The plots do not have labelled x and y-axes.
* The legends are very confusing. As an example: "on the left (respectively, on the right)."

There are also numerous prior works around deep kernel processes and machines that would be worth discussing in the related work:
* https://arxiv.org/abs/2010.01590
* https://arxiv.org/abs/2108.13097

while this work uses a very different theoretical approach, it ultimately addresses similar conceptual issues.

**Questions:**

N/A

---

> ### Author Response · Authors · 2024-11-19
>
> Comment: The experimental results are presented poorly. .
>
> Answer: We have improved the presentation of our experimental results. Please see Figures 1 and 2 in our revised submission.
>
> Comment: There are numerous prior works around deep kernel processes and machines that would be worth discussing in the related work. While this work uses a very different theoretical approach, it ultimately addresses similar conceptual issues.
>
> Answer: We thank the reviewer for sharing these references which we have incorporated into the related work discussion. Please see l. 47-49 in our revised submission.

---

> > ### Comment · Reviewer_U5Zu · 2024-11-26
> >
> > Thanks!  I have raised my score.

---

### Official Review · Reviewer_LMnA · 2024-11-04

**Soundness:** 2
**Presentation:** 2
**Contribution:** 2
**Rating:** 5
**Confidence:** 3

**Summary:**

This paper studies bayesian neural networks in the linear ($P/N = \text{constant}$ for data $P$ and width $N$) regime and sub-linear ($P/(N N_0) = \text{constant}$) regime. The authors operate under a spectral universality assumption, which treats the eigenfunctions of the limiting kernel as random with a covariance determined by the posterior kernel. The authors argue that this spectral universality assumption is logically equivalent to a kernel renormalization theory which was derived for deep linear networks. This kernel renormalization is a scale shift in the kernel by a variable $u$ which needs to be solved for self-consistently. In the sublinear regime, the kernels are rank-deficient which the authors acknowledge by allowing for a singular spectrum and utilizing the pseudo-inverse.

**Strengths:**

This paper studies the important problem of characterizing feature learning in nonlinear Bayesian neural networks and provides an original idea to apply a spectral universality idea to characterize feature learning. It further aims to justify some of the recent applications of kernel renormalization theory to nonlinear networks.  Showing that the spectral universality assumption implies and is implied by the kernel renormalization picture is an interesting contribution. The authors also provide a few experiments to support their claims.

**Weaknesses:**

However, the spectral universality assumption is not proven directly and the distribution of kernel eigenfunctions in either the proportional regime or the sublinear regime has not been characterized outside of the spectral universality assumption. The experiments are somewhat limited. I have a number of questions below, which if addressed could lead me to increase my score.

**Questions:**

1. What do the authors suspect would happen for networks in the $N \gg P$ regime with mean-field / $\mu$P scaling that preserves strong feature learning? In this case, [theory](https://arxiv.org/abs/2205.09653) predicts that the kernels are non-random but the preactivations can become non-Gaussian as was also shown in the [Bayesian setting](https://arxiv.org/abs/2406.16689). Do the spectral universality assumptions break down?
2. What do the authors think of the spectral universality assumption for cases where the matrices [pick up low rank spikes](https://arxiv.org/abs/2410.18938) such as learning single or multi-index models.  Do spiked kernels violate the kernel renormalization theory?

---

> ### Author Response · Authors · 2024-11-19
>
> Question: What do the authors suspect would happen for networks in the $N >>P$ regime with mean-field $\mu P$ scaling that preserves strong feature learning? In this case, theory predicts that the kernels are non-random but the preactivations can become non-Gaussian as was also shown in the Bayesian setting. Do the spectral universality assumptions break down?
>
> Answer: The spectral universality assumption relates to the prior kernel, not the posterior (as discussed l. 317-332 in our initial submission and l. 328-338 in our revised submission). However, it plays a crucial role in calculating the posterior by enabling integration over the prior. In the light of this, we do not think that the spectral universality assumption would cease to apply with the (non-lazy) scaling. This observation is consistent with the calculations from van Meegen and Sompolinsky (2024) in their appendix (equations A68 and A69), where they establish that the mean predictor remains unchanged compared to the lazy scaling (which is the scaling that we have) and that the variance is renormalised by a different factor (which vanishes as $N \to \infty$). This is exactly the behavior we would expect if the spectral universality assumption held, as hinted by our Theorem 3.5. Furthermore, the renormalisation factor is highly dependent on the limiting spectral measure (in the lazy scaling, our Marchenko-Pastur map) as established by the proof of our Theorem 3.5.
>
>
> Comment: What do the authors think of the spectral universality assumption for cases where the matrices pick up low rank spikes such as learning single or multi-index models. Do spiked kernels violate the kernel renormalization theory?
>
> Answer: In the aforementioned paper, the authors demonstrated an equivalence between training a two-layer neural network using only the first gradient step and training a low-rank spiked random feature model. This can be interpreted as altering the (prior) data representation to reside in a low-dimensional space, which we believe significantly constrains the orthogonal matrices permissible for constructing the prior kernel. As a result, we anticipate that the spectral universality assumption would fail in this scenario. However, this does not impact the validity of the spectral universality assumption in the contexts considered in our paper, as a single gradient descent step is not expected to replicate the full Bayesian training of a network.

---

> ### Author Response · Authors · 2024-11-21
>
> The line numbering has changed again in our new revised submission (in response to a feedback from Reviewer H2W1). Please see below the line numbers relevant to your review.
>
> "l. 317-332 in our initial submission and l. 328-338 in our revised submission" ->  l.332-342 in our new revised submission

---

> ### Comment · Reviewer_LMnA · 2024-11-22
>
> I thank the authors for answering my questions and clarifying my confusion regarding posterior/prior kernel. I will revise my score. My one remaining recommendation would be to emphasize that the current theory may not be able to capture the spiked kernels which result from large steps of feature learning, which would break the universality assumption.
>
> Additionally, in the appendix, I think the variance should be $\left< (\delta f)^2 \right>$ rather than $\left< \delta f \right>$, unless I am mistaken.

---

> > ### Author Response · Authors · 2024-11-22
> >
> > Thank you for your suggestion. We have added this insight to the paper (see l.389-394 in our revised submission). And thank you for spotting this error in the notation, we have corrected it.

---

### Meta-Review · Area_Chair_2XV9 · 2024-12-20

**Metareview:**

This paper investigates Bayesian neural networks (BNNs) in certain asymptotic limits, drawing connections between kernel-theoretic approaches and statistical mechanics. It presents novel integral formulas for BNN predictors in linear and sublinear width regimes and extends renormalization theory, originally developed for linear networks, to nonlinear BNNs. Reviewers point to weaknesses in the  empirical validation, noting that the experiments are limited to simple datasets and lack detailed explanation. Furthermore, the core spectral universality assumption remains unproven, and the practical applicability of the derived integral formulas is not entirely clear due to computational concerns. Despite these limitations, the theoretical contributions are strong and original. The paper attacks the crucial problem of characterizing feature learning in nonlinear BNNs and offers a novel approach by combining spectral universality with kernel renormalization. The derivation of integral formulas for BNN predictors in different width regimes and the extension of renormalization theory represent useful advancements in the theoretical understanding of BNNs, and therefore I recommend publication.

**Additional Comments On Reviewer Discussion:**

The reviewers raised several questions regarding the spectral universality assumption, its applicability in different scaling regimes or with spiked kernels, and its relation to kernel renormalization theory. The authors clarified that the assumption applies to the prior kernel and is crucial for integrating over the prior to calculate the posterior, and they believe it holds in the non-lazy scaling regime, supported by calculations in related work. However, they acknowledge it likely fails when low-rank structures are introduced, as in spiked kernels or single-step gradient descent training.

Reviewers also questioned the experimental results, their presentation, and computational feasibility. The authors addressed these concerns by improving the presentation of the figures, providing details about the experimental setup, and explaining the computational challenges and limitations of working with finite-size systems despite the theoretical results being derived for infinite limits. They also addressed questions regarding the notation and clarified the applicability of their approach to other activation functions beyond ReLU.

Overall, the discussion phase resolved many of the reviewers' questions and concerns, including those of Reviewer LMnA, who intended to increase their score (but presumably forgot to do so). It is worth pointing out that the recommendation for acceptance does not hinge on this score increase, as the issues at question do not seem crucial for establishing the overall importance of the paper.

---

### Decision · Program_Chairs · 2025-01-22

Accept (Poster)